# Implementation of Physical Activity Programs for Rural Cancer Survivors: Challenges and Opportunities

**DOI:** 10.3390/ijerph182412909

**Published:** 2021-12-07

**Authors:** Kelly A. Hirko, Joan M. Dorn, James W. Dearing, Catherine M. Alfano, Annemarie Wigton, Kathryn H. Schmitz

**Affiliations:** 1Department of Epidemiology and Biostatistics, College of Human Medicine, Michigan State University, East Lansing, MI 48824, USA; 2Department of Exercise and Nutrition Sciences, School of Public Health and Health Professions, University at Buffalo, Buffalo, NY 14214, USA; joandorn14@gmail.com; 3Department of Communications, Michigan State University, East Lansing, MI 48824, USA; dearjim@msu.edu; 4Northwell Health Cancer Institute, New Hyde Park, NY 11042, USA; calfano3@northwell.edu; 5Munson Healthcare, Traverse City, MI 49684, USA; annemariewigton@gmail.com; 6Department of Public Health Sciences, Penn State College of Medicine, Hershey, PA 17033, USA; kschmitz@phs.psu.edu

**Keywords:** physical activity, rural, cancer, implementation, health equity, translation

## Abstract

Physical activity after cancer diagnosis has been consistently associated with improvements in quality of life and prognosis. However, few cancer survivors meet physical activity recommendations, and adherence is even lower among those living in rural settings. The purpose of this quasi-experimental study was to evaluate the implementation of a clinic-based physical activity program for cancer survivors at a rural community oncology setting. We also examined changes in quality-of-life measures among 24 cancer survivors participating in the physical activity program and described challenges and opportunities to optimize future implementation efforts in rural settings. Significant pre- to post-program improvements in fatigue (5.5 to 6.8; *p* = 0.03), constipation (7.7 to 9.0; *p* = 0.02), pain (6.7 to 8.0; *p* = 0.007), and sleep quality (*p* = 0.008) were observed. Participants also reported improved nausea, stamina, depression, stress, and overall physical health after participation in the physical activity program, although the differences were not statistically significant (all *p*-values > 0.13). However, the reach of the physical activity program was limited, with only 0.59% of cancer survivors participating. Fidelity to the physical activity program was relatively high, with 72.7% of survivors participating in at least five classes. Our findings suggest that physical activity programs in oncological settings may need tailoring to effectively reach rural cancer survivors.

## 1. Introduction

Physical activity after cancer diagnosis has been consistently linked with improvements in multiple cancer-related health outcomes, including fatigue, quality of life, physical functioning, and overall prognosis [1,2]. Indeed, results from a recent pooled analysis of 26 observational studies suggest a 37% lower risk of dying from cancer among individuals engaging in higher levels of physical activity, with consistent risk reductions observed across breast, colorectal, and prostate cancer survivors [2]. Supported by this growing body of evidence demonstrating the benefits of physical activity in cancer survivorship, several national and international organizations have published clinical guidelines recommending regular exercise for individuals diagnosed with cancer [1,3,4,5]. However, few cancer survivors meet current physical activity guidelines for health (at least 150 min/week of moderate aerobic exercise and 2 or more days/week strength training) [6], and adherence is even lower among those living in rural settings [7,8].

Nearly 20% of Americans and an estimated 21% of cancer survivors reside in rural regions [9]. Rural areas are often underrepresented in cancer research [10], and are one of the largest medically underserved groups of cancer survivors in the U.S. [11]. Compared to their urban counterparts, rural cancer survivors are more likely to be physically inactive, sedentary, and obese [12,13,14], and are more likely to die from cancer [15]. Rural cancer patients face many challenges in accessing care, including limited availability of cancer supportive care providers, transportation barriers, financial issues, and limited access to clinical trials [16]. As such, physical activity interventions that work in urban populations may need tailoring to be effective for rural cancer survivors. Thus, identifying effective strategies to increase physical activity among rural cancer survivors is urgently needed.

The purpose of this study was to evaluate the implementation of a clinic-based physical activity program for cancer survivors at a rural community oncology setting. We also examined changes in validated quality of life measures among cancer survivors participating in the physical activity program and describe challenges and opportunities to optimize future implementation efforts of physical activity programs in rural oncological settings.

## 2. Materials and Methods

### 2.1. Study Population

This quasi-experimental study was conducted at Cowell Family Cancer Center/Munson Medical Center (CFCC/MMC), a regional cancer center located in rural Northwest Michigan. All of the counties served by CFCC/MMC are considered rural based on the Economic Research Services’ rural-urban continuum codes. Cancer patients treated at CFCC/MMC were able to participate in a clinic-based physical activity program led by oncology-certified physical trainers at any point during their cancer care continuum. Cancer patients were provided information about these programs at diagnosis and were also referred to the program by cancer support services staff at various points throughout their care. Cancer patients participating in the clinic-based physical activity program between May 2017 and September 2019 were eligible to participate in this study. All cancer survivors participating in the physical activity program were informed of the research study and provided informed consent. Eligible cancer survivors were not randomized to participate in the physical activity program, and only participants who attended at least five classes were evaluated. This study was approved by the Institutional Review Board at Munson Medical Center (Registration #00002661).

### 2.2. Physical Activity Program

The physical activity program in this study sought to increase progression toward achieving the physical activity guidelines [6], recognizing that the physical activity program must be appropriate for cancer survivors’ level of overall health, ability, and comorbidities. The physical activity program was designed for cancer patients during all stages of treatment and survivorship and complied with the American College of Sports Medicine recommendations for 2–3 sets of resistance exercises at a weight that can be performed for 8–12 repetitions [17]. Each small group class was approximately 45 min in length and consisted of a dynamic warm up, followed by strength and aerobic training using free weights to apply resistance focused on muscles of the chest, back, shoulders, quadriceps, hamstrings, gluteals, biceps and triceps. Aerobic training combined high- and low-impact floor exercises designed to keep the participant moving and increase heart rate with moderate to vigorous intensity. All training exercises could be modified for lower intensity, particularly for novice participants. The physical activity program was led and supervised by oncology-certified physical trainers. Participants were encouraged to participate in weekly physical activity sessions, but the program allowed flexibility in scheduling future sessions. There was no cost for cancer survivors to participate in the clinic-based physical activity program.

### 2.3. Quality of Life Assessment

Quality of life was assessed using an abbreviated form of the Quality of Life Patient/Cancer Survivor Version (QOL-CSV) instrument, which is based on previous versions developed by researchers at the City of Hope National Medical Center [18]. This validated instrument was revised in cancer survivorship studies and includes 41 items representing the four domains of quality of life, including physical wellbeing, psychological wellbeing, social wellbeing, and spiritual wellbeing. The overall re-test reliability for this instrument was 0.89 and the internal consistency using Cronbach’s alpha coefficient was r = 0.93 [19,20]. For this study, we included the following eight QOL dimensions: fatigue, constipation, nausea, stamina, anxiety, depression, overall stress, and overall pain. Participants also self-reported their overall physical health (ranging from extremely poor to excellent). The scoring was based on a scale of 0 (worst outcome) to 10 (best outcome), with reverse anchors used for fatigue, constipation, nausea, anxiety, depression, overall stress, and overall pain. Using specific questions from the Pittsburgh Sleep Quality Index [21], participants also reported perceived sleep quality (very good, fairly good, fairly bad, very bad) and the hours of actual sleep on most nights during the past month. Participants were asked to answer all questions based on their life at the time of the completion of the questionnaire, and to circle the number that best describes their experiences. Self-administered questionnaires were provided to study participants upon arrival to the group class and were collected by the oncology-certified physical trainer. Questionnaires were administered at baseline (before participation in the initial physical activity program), and after completion of every five physical activity sessions. Data from the research questionnaire responses were entered into a spreadsheet for analysis.

### 2.4. Implementation Outcomes

The primary outcome of this study was the overall physical activity program implementation. The implementation outcomes assessed in this study were based on the Reach, Effectiveness, Adoption, Implementation, and Maintenance (RE-AIM) framework [22,23], which can be applied to assist with the translation of research to practice and to estimate the public health impact of programs and interventions. In this study, we assessed the reach of the physical activity program, defined as the proportion of eligible cancer survivors treated at CFCC/MMC over the study period, ascertained from hospital reports, and who participated in the physical activity program. The effectiveness of the program was evaluated by the change in QOL metrics pre-post participation in the physical activity program. We also assessed the implementation fidelity at the individual-level by examining participants’ use of the physical activity program, including adherence to the intervention (the proportion of cancer survivors participating in at least five physical activity classes) and dose of the intervention (the number of classes attended, categorized as low (5 classes), moderate (10 classes), and high (≥15 classes) [24].

### 2.5. Statistical Analysis

Descriptive analyses were used to evaluate study participant characteristics (age, sex, and cancer site). Continuous variables were summarized with means and standard deviations (or medians and interquartile range) to assess homogeneity. We visually inspected histograms to ensure a normal distribution of continuous variables. We calculated differences in the scores for QOL metrics, sleep quantity, and quality and physical functioning before and after participation in the physical activity program using paired samples *t*-tests. Post-program assessments were calculated from the last available date of participation in the physical activity program. In sensitivity analyses, we evaluated whether differences in the scores for QOL metrics and physical functioning before and after participation in the physical activity program differed according to the dose of physical activity. All statistical analyses were performed using SAS, version 9.4 (SAS Institute, Cary, NC, USA); statistical significance was defined at *p* ≤ 0.05.

## 3. Results

### 3.1. Study Characteristics at Baseline

Of the 35 cancer survivors participating in the onsite physical activity program, 33 (94.3%) agreed to participate in the study and completed the baseline questionnaire. Among those enrolled in the study, 24 (72.7%) participated in at least five physical activity classes and were included in the analysis of change in QOL metrics from before the first class to the end of the last class (ranged from 5 to 50 classes). As shown in Table 1, participants ranged in age from 31 to 72 years, with a median age of 66 years, and interquartile range of 12.5 years. The majority of study participants were female (75.0%) and had been diagnosed with breast cancer (33.3%), followed by gastrointestinal cancer (12.5%), gynecologic cancer (12.5%), prostate cancer (12.5%), and skin cancer (8.3%).

### 3.2. Program Implementation

Only 33 cancer survivors participated in the physical activity program over the 29-month enrollment period. During this study period, a total of 5605 cancer patients were treated at CFCC/MMC. As such, the reach of the physical activity program was limited, with only 0.59% of cancer survivors participating. Of the 33 participants enrolled in the physical activity program, 24 (72.7%) attended at least five classes, suggesting program adherence. Of these participants, 7 (29.2%) completed 5 classes (low dose), 7 (29.2%) completed 10 classes (moderate dose), and 10 (41.7%) completed 15 or more classes (high dose), with an average of 12 physical activity classes attended per participant (range, 5–45 classes).

Self-reported QOL measures pre- and post-participation in the physical activity program are shown in Figure 1. Participants reported worse scores for stamina, fatigue, and overall physical health, and better scores with regard to nausea both pre- and post-program participation. Clinically meaningful and statistically significant improvements in fatigue (5.5 to 6.8; *p* = 0.03), constipation (7.7 to 9.0; *p* = 0.02), and pain (6.7 to 8.0; *p* = 0.007) were observed. Participants also reported improved nausea, stamina, depression, stress, and overall physical health after participation in the physical activity program, although the differences were not statistically significant (all *p*-values > 0.13). Anxiety was the only metric that did not show improvement after participation in the physical activity program. In sensitivity analyses, the observed improvements in QOL metrics were generally similar regardless of physical activity dose, although the only statistically significant improvements were observed for fatigue (5.0 to 7.3; *p* = 0.02) and pain (6.3 to 7.9; *p* = 0.02) among those receiving a high dose.

### 3.3. Changes in Sleep

At baseline, participants reported an average of 6.8 h (range: 4–11) of actual sleep on most nights during the past month. Post-participation in the physical activity program, the reported hours of sleep on most nights slightly increased to an average of 7.0 h (range 4–10), but the difference was not statistically significant (*p* = 0.49). However, participants reported significantly improved sleep quality after participation in the physical activity program (*p* = 0.008). For example, 41.7% of participants reported fairly bad sleep quality at the baseline, whereas only 8.3% still reported fairly bad sleep quality after participation in the physical activity program. Likewise, the percentage of participants reporting very good sleep quality increased from 12.5% to 25.0% after participation in the physical activity program.

## 4. Discussion

In this study of rural cancer survivors, significant improvements in cancer-related fatigue, constipation, pain, and sleep quality were observed after participation in a clinic-based physical activity program. However, overall participation in the clinic-based physical activity program was limited, suggesting that physical activity interventions may need tailoring to effectively reach rural cancer survivors. As such, consideration of unique barriers and preferences for physical activity in rural settings is needed to optimize the feasibility and uptake of physical activity interventions for rural cancer survivors.

Findings from this study provide further evidence to support the effectiveness of physical activity for improving cancer-related quality of life metrics, including fatigue, constipation, and pain [1]. Improvements in these QOL metrics were noted despite lower levels of physical activity than observed in most prior studies citing benefits (i.e., 3 sessions/week for 30–60 min per session of moderate to vigorous physical activity) [1]. Importantly, this study adds to emerging evidence suggesting the benefits of physical activity on sleep quantity and quality among cancer survivors. Although consistent associations between physical activity and sleep quality have been observed in prior epidemiologic studies [25,26], data on cancer survivors specifically are mixed. For example, results from a meta-analysis of breast cancer survivors suggested no benefits of exercise interventions on sleep quality [27], whereas improvements were observed in a recent randomized controlled trial of breast cancer survivors [28]. Given that sleep quality is recognized among the top five highest ranked patient-reported outcomes of importance [29], further studies to characterize the role of physical activity in sleep quality among cancer survivors are warranted.

The clinic-based physical activity program evaluated in this study did not reach the vast majority of cancer survivors in this rural setting. This finding may reflect both the poor overall implementation of the physical activity program into clinical pathways and procedures, and the limitations of a clinic-based program in a geographically dispersed rural region. Indeed, we identified several key issues related to the low participation in this clinic-based physical activity program and discuss these issues in the proceeding paragraphs.

First, this clinic-based physical activity program was implemented without directly engaging oncology providers. The effectiveness of oncology provider referrals for increasing physical activity among cancer survivors has been demonstrated in several randomized studies [30,31,32,33]. As such, the Moving Through Cancer initiative of the American College of Sports Medicine advocates for clinicians to assess, advise, and refer oncology patients to appropriate exercise programming resources [34]. However, there is limited uptake of these recommendations in practice, with oncology providers citing primary barriers for providing physical activity referrals related to lack of education, time, and appropriate referral programs [35,36]. The lack of supportive infrastructure, including appropriate referrals or effective strategies to enhance discussion of physical activity in the clinical setting, may also inhibit oncology provider referrals. Addressing these barriers is critical to increasing oncology provider referrals and uptake to physical activity programs in oncology settings.

Second, we did not assess barriers to participation in a clinic-based physical activity program for rural cancer survivors prior to implementation. As in many cancer centers serving patients in geographically dispersed rural regions, the oncology clinic in this study serves a 30-county region, with patients travelling an average of 60 miles round trip to receive cancer care. Given the inherent transportation-related challenges in rural settings [16], offering only a clinic-based physical activity program may not be acceptable nor feasible for rural cancer survivors. Indeed, home-based physical activity interventions have been linked to greater reach and reduced participant burden [33], and may be necessary to increase reach and potential sustainability in rural settings.

Strengths of this study include the focus on rural cancer survivors, a generally understudied population in cancer survivorship research. However, this study population was small, and we were unable to specifically assess whether the effectiveness of the physical activity program differed according to cancer site, treatment course, and other participant characteristics. Given the low participation in the physical activity program, this study population may not be representative of all rural cancer survivors. Indeed, study participants included survivors of multiple cancer types with varying medical and surgical treatments, which could have differentially impacted physical activity program effectiveness. Additionally, we did not have information on baseline physical activity levels, treatment type or timing with regard to participation in the physical activity program, distance travelled to the cancer center, and other potential confounders which could have affected program effectiveness and implementation outcomes. Finally, this was not a randomized study, and we did not have a control group to assess potential changes in QOL metrics over the same time period, which may have been unrelated to the physical activity program.

Improving adherence to physical activity guidelines among rural cancer survivors requires access to acceptable physical activity programs that have been evaluated in rural populations. Prior research on physical activity in cancer survivors has largely focused on cumbersome, expensive, and strictly-controlled physical activity regimens within randomized study settings; however, these approaches are unlikely to work in real-world community oncology settings [37]. The need for research to understand how to effectively translate physical activity research into clinical and community oncology practice has been recognized as a critical research question to advance the field of physical activity and cancer survivorship [38]. Findings from this study underscore the importance of considering context when implementing evidence-based physical activity interventions, and suggest that rural community oncology clinics may require different approaches to increase physical activity from those used in academic hospital settings.

## 5. Conclusions

The successful implementation of physical activity programs in rural community oncology settings will require tailoring to address specific needs and preferences of cancer survivors, oncology providers, and the rural community oncology setting. A one-size-fits-all approach is not likely to work, and cancer survivors may need access to a variety of safe, accessible, and effective physical activity programs to increase uptake. Importantly, the environmental, policy, and system-level barriers, which provide minimal support for being physically active [39], particularly in rural settings, need to be addressed to increase sustained physical activity among rural cancer survivors. As such, a socio-ecological approach using multilevel interventions designed to work synergistically to overcome limitations of strategies targeting any single level of influence has the greatest potential to improve physical activity on a population level [40,41,42]. Moreover, there is a critical need to increase oncology provider competencies, develop tools to assist providers, and build capacity in clinical processes and workflow to increase the likelihood of oncology provider referrals to appropriate programs aligned with survivors’ needs and preferences. Advancing the implementation of physical activity programs for rural cancer survivors will also require comprehensive evaluation of the reach, effectiveness, adoption, implementation, and maintenance (individual and organizational) of physical activity interventions in accordance with the RE-AIM framework. Additional studies are also needed to identify effective strategies to integrate evidence-based physical activity programs into standard cancer care, particularly in community oncology settings (outside of NCI-designated cancer centers) where the majority of cancer care occurs.

## Figures and Tables

**Figure 1 ijerph-18-12909-f001:**
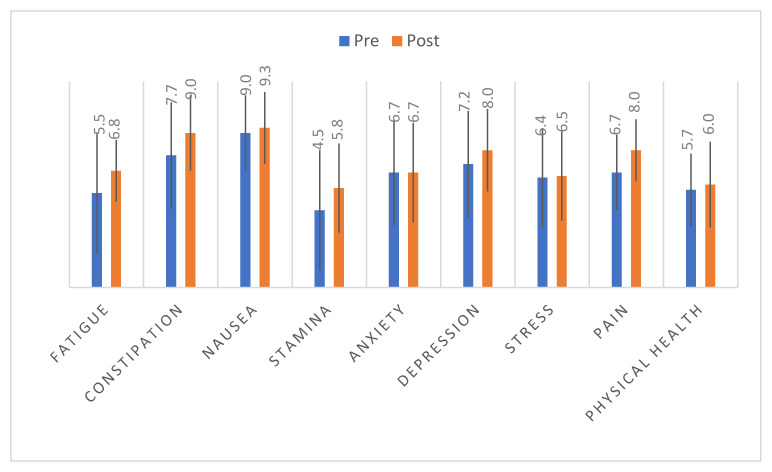
Mean score (0 = worst outcome to 10 = best outcome) and standard deviation bars for pre- and post-participation in the PA program.

**Table 1 ijerph-18-12909-t001:** Characteristics of participants in the physical activity program (*n* = 24).

Characteristic	Median (IQR) or *N* (%)
Age	66.0 (12.5)
Female	18 (75.0)
Cancer type	
Breast	8 (33.3)
Gastrointestinal	3 (12.5)
Gynecologic	3 (12.5)
Prostate	3 (12.5)
Lung	1 (4.2)
Kidney	1 (4.2)
Skin	2 (8.3)
Multiple myeloma	1 (4.2)
Sarcoma	1 (4.2)
Missing	1 (3.0)

## Data Availability

The data presented in this study are available on request from the corresponding author. The data are not publicly available due to the agreements set forth in the informed consent document.

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
