# Peer review of "Implementation of Physical Activity Programs for Rural Cancer Survivors: Challenges and Opportunities"

_ijerph, 2021, doi:10.3390/ijerph182412909_

Round 1

Reviewer 1 Report

The new version of the manuscript titled “Implementation of Physical Activity Programs for Rural Cancer Survivors: Challenges and Opportunities” has incorporated suggestions of changes made to the previous version.

The new version explicitly states the study design as a quasi-experimental study at the beginning of the materials and methods section.

The new version of the manuscript also has important improvements in its discussion section. In fact, the discussion section has been reorganised and limitations of the study have been highlighted as well as strengths of the research.

The new version of the manuscript has enhanced consistency among its different sections and gained robustness.

Author Response

Thank you for your comments.   We agree that this revised manuscript has enhanced consistency and gained robustness. 

Reviewer 2 Report

Overall, the introduction to this paper has convinced me very much that it is an interesting topic with a high need for research, so that I will recommend the paper as a whole with major revisions. Unfortunately, this paper adds little new approaches and information to the research question - it is particularly a pity that no barriers to implementation in rural settings have been identified. Some revisions should be made, especially the method part and the result part, in order to increase the quality and science of the paper.

Abstract:
- Please state the number of patients who participated in the study and please also go into more detail about the study design.

Material and methods:
- please state explicitly that no randomization was carried out and that only subjects who had gone through the program were tested.
- Please state exactly which tests were carried out at which times
- Please state the primary outcome of this study
- Are there any costs to participate?
- How long after the cancer diagnosis were the patients locked in?
- please include the number of the ethics committee vote
- Please provide information about the registration of the study and give the registration number
- Further down in the discussion part it is stated that the study was "underpowered". However, a power calculation cannot be found in the methods, please add, in particular how big the sample size planning was
- Statistical analyzes: Please state whether a test for normal distribution has been carried out beforehand and please also state that the median IQR and the mean standard deviation were stated.

Results:
- The part of the results should consistently be reformulated in a purely objective manner, there are evaluations here in many places (cf. line 193) which do not belong in this section.
- The distribution of age does not seem to me to be normally distributed, so that the median should certainly be given here instead of a mean
- The time from the cancer diagnosis and the time until the start of the program would certainly be an important addition
- Please add a table to the graphic, which clearly shows the times of the tests, the test results and the changes in the course

Discussion:
- One of the major limitations of the study is certainly that there was no control group. Thus, in particular, all changes in the tests that have been reported are to be viewed with reservations, as one cannot rule out that these changes may have occurred "by themselves" with sufficient distance to the cancer diagnosis and therapy. At this point it would be interesting to make a comparison in the discussion with studies that carried out a randomization with the question of whether the "intervention group" presented here with the control groups there differs from the results.
- In my opinion, the most important outcome of this study is that it was just as effective as in the literature, but that the uptake was extremely poor. In my opinion, this should also be mentioned directly above in the discussion. It would therefore be important in the discussion to put the reasons for the poor utilization in the foreground. In the method section, more information is required on the location / environment / information for patients / advertising / registration modalities / costs etc.

Author Response

Reviewer #2: Overall, the introduction to this paper has convinced me very much that it is an interesting topic with a high need for research, so that I will recommend the paper as a whole with major revisions. Unfortunately, this paper adds little new approaches and information to the research question - it is particularly a pity that no barriers to implementation in rural settings have been identified. Some revisions should be made, especially the method part and the result part, in order to increase the quality and science of the paper.

We are happy to hear that the reviewer recognizes the importance and relevance of this research topic.  We agree that it would be helpful to describe barriers to implementation of physical activity programs in rural settings, but this information was not systematically collected as part of this study.  However, we do describe specific barriers for rural cancer patients identified in prior studies as follows “Rural cancer patients face many challenges in accessing care, including limited availability of cancer supportive care providers, transportation barriers, financial issues, and limited access to clinical trials” (lines 62-65).  Moreover, we discuss several key issues related to the low participation in this physical activity program in the 4th and 5th paragraph of the Discussion section, including not directly engaging with oncology providers to provide referrals to the physical activity program and not specifically assessing barriers to participation in a clinic-based physical activity program for rural cancer survivors prior to implementation.  Taken together, we hope that this brief report will be informative for other oncology programs seeking to implement physical activity programs for rural and other underserved cancer survivors.  We have addressed the reviewer’s additional suggestions to revise methods and results in our responses below.

Abstract:
- Please state the number of patients who participated in the study and please also go into more detail about the study design.

Thank you for this helpful suggestion.  We have included the number of participants and the study design in the revised abstract.

Material and methods:
- please state explicitly that no randomization was carried out and that only subjects who had gone through the program were tested.

We have added the following text in the Methods, section 2.1 Study Population to address this point; “Eligible cancer survivors were not randomized to participate in the physical activity program, and only participants who attended at least five classes were evaluated.”

- Please state exactly which tests were carried out at which times

Thank you for noting this important clarification.  We described the specific quality of life assessment tools in section 2.3.  We have added the following text on lines 141-144 to clarify timing of these assessments, “Questionnaires were administered at baseline (before participation in the initial physical activity program), and after completion of every five physical activity sessions.

- Please state the primary outcome of this study

Thank you for this comment.  The primary outcome of the study was the overall program implementation, assessed using the RE-AIM framework and described in section 2.4.  This includes reach of the program, effectiveness of the physical activity program on quality of life metrics and fidelity to the physical activity program.  To clarify this point, we have added the following sentence in section 2.4, “The primary outcome of this study was the overall physical activity program implementation”.

- Are there any costs to participate?

There was no cost for cancer survivors to participate in the clinic-based physical activity program.  We have added this information in section 2.2 where we describe the physical activity program.

- How long after the cancer diagnosis were the patients locked in?

As described in section 2.1 of the manuscript, all patients treated for cancer at the Cowell Family Cancer Center/Munson Medical Center (CFCC/MMC) were eligible to participate in the physical activity program.  Participants were provided information about the physical activity program at diagnosis, and were also referred to the program by cancer support services staff at various points throughout their care.  Thus, there was variation in terms of timing of participation with regard to cancer diagnosis date.  To clarify this point, we have added the following text in section 2.1, “Cancer patients treated at CFCC/MMC were able to participate in a clinic-based physical activity program led by oncology-certified physical trainers at any point during their cancer care continuum.”

- please include the number of the ethics committee vote

This study was approved by the Munson Healthcare Institutional Review Board (Registration #00002661).  We have updated the text on page 2, section 2.1 with this information. 

- Please provide information about the registration of the study and give the registration number

This study was approved by the Munson Healthcare Institutional Review Board (Registration #00002661).  We have updated the text on page 2, section 2.1 with this information.  Given that this study was not a clinical trial, we did not register the study with ClinicalTrials.gov

- Further down in the discussion part it is stated that the study was "underpowered". However, a power calculation cannot be found in the methods, please add, in particular how big the sample size planning was

Thank you for addressing this point.  This study was not specifically hypothesis-driven and therefore, no power calculation was conducted.  Rather, the purpose of this study and brief report was to describe the overall implementation of a clinic-based physical activity program in a rural oncology setting.  We did not have a specific goal in terms of study population size.  To avoid confusion, we have revised the text in the Discussion, removing the word “underpowered” on line 312.  The revised  sentence now reads, “However, this study population was small and we were unable to specifically assess whether the effectiveness of the physical activity program differed according to cancer site, treatment course and other participant characteristics.”

- Statistical analyzes: Please state whether a test for normal distribution has been carried out beforehand and please also state that the median IQR and the mean standard deviation were stated.

As stated in the Statistical Analysis subsection 2.5, “continuous variables were summarized with means and standard deviations (or medians) to assess homogeneity.  We visually inspected histograms to ensure normal distribution of continuous variables.”  To address the reviewer’s point, we have added description of specific IQR assessment when evaluating normality of continuous variable distribution as follows: “Continuous variables were summarized with means and standard deviations (or medians and interquartile range) to assess homogeneity.

Results:
- The part of the results should consistently be reformulated in a purely objective manner, there are evaluations here in many places (cf. line 193) which do not belong in this section.

We appreciate the reviewer’s comment.  We reviewed the results section in detail to ensure that findings are presented in a purely objective manner.  The reviewer specifically points to text on line 193 which states the number of cancer patients treated at the study site during the enrollment period, which was used to calculate program reach.  This information was objectively collected from the registered cancer diagnoses at the study site (CFCC/MMC).  Given the reviewer’s concern we have revised the following text in section 2.4 to clarify this assessment of program reach; “In this study, we assessed reach of the physical activity program, defined as the proportion of eligible cancer survivors treated at CFCC/MMC over the study period, ascertained from hospital reports, who participated in the physical activity program.” 

All of the other information on program participation numbers presented in the results section were documented objectively in a data spreadsheet described in the text (lines 139-141).

- The distribution of age does not seem to me to be normally distributed, so that the median should certainly be given here instead of a mean

Thank you for this comment.  We have updated the results text in section 3.1 and Table 1 to include the median and IQR for age as follows; “As shown in Table 1, participants ranged in age from 31 to 72 years, with a median of 66 years, and interquartile range of 12.5 years.

- The time from the cancer diagnosis and the time until the start of the program would certainly be an important addition

We agree with the reviewer that this information on timing of physical activity participation with regard to cancer diagnosis would be informative.  However, given that our focus was on the implementation of the physical activity program in a rural oncology setting and not specifically on program effectiveness, we did not collect this information as part of the study.  Prior studies focused on effectiveness of physical activity in cancer survivorship have addressed issues related to timing of physical activity in the cancer care continuum, but this was beyond the scope of this paper.

- Please add a table to the graphic, which clearly shows the times of the tests, the test results and the changes in the course

We apologize for the confusion.  All of the baseline (pre-participation) assessments were conducted before individuals participated in the first physical activity program class.  This is described in the methods, including the following additional text added in this revision for clarification; “Questionnaires were administered at baseline (before participation in the initial physical activity program), and after completion of every five physical activity sessions.”  The test results (scores) are shown in the figure for each metric at both the pre- and post- assessment timepoint.  The scoring for each metric is described in detail in section 2.3 of the Methods.  The changes in scores pre- and post-participation in the physical activity program are captured in the bar graph figure by visually comparing the blue vs. orange bars.

We describe the number of participants who completed the baseline questionnaire and each additional post-participation questionnaire in the Results section 3.2.  Additionally, in sensitivity analysis results described on lines 235-238, we report improvements in QOL metrics based on dose (or number of physical activity classes that survivors attended).

Discussion:
- One of the major limitations of the study is certainly that there was no control group. Thus, in particular, all changes in the tests that have been reported are to be viewed with reservations, as one cannot rule out that these changes may have occurred "by themselves" with sufficient distance to the cancer diagnosis and therapy. At this point it would be interesting to make a comparison in the discussion with studies that carried out a randomization with the question of whether the "intervention group" presented here with the control groups there differs from the results.

We agree with the reviewer that the lack of control group is a limitation of the analysis of program effectiveness in this study.  We have acknowledged this in the Discussion section in this revision as follows, “Finally, this was not a randomized study and we did not have a control group to assess potential changes in QOL metrics over the same time period, which may have been unrelated to the physical activity program.”  Moreover, we have acknowledged that “this study population may not be representative of all rural cancer survivors” and certainly of all cancer survivors participating in prior randomized studies of physical activity in cancer survivorship.  However, given that the evidence base for the effectiveness of physical activity on cancer outcomes has been established in multiple prior large epidemiologic studies, the main goal of this study was to evaluate the implementation of a clinic-based physical activity program in a real-world rural oncology setting.  Thus, we do not believe that the lack of a control group limits the generalizability of our findings with regard to implementing physical activity programs in rural oncology settings.

- In my opinion, the most important outcome of this study is that it was just as effective as in the literature, but that the uptake was extremely poor. In my opinion, this should also be mentioned directly above in the discussion. It would therefore be important in the discussion to put the reasons for the poor utilization in the foreground. In the method section, more information is required on the location / environment / information for patients / advertising / registration modalities / costs etc.

We completely agree with the reviewer that the important finding from this study is the overall low participation in clinic-based physical activity program in this rural oncology setting.  We discuss this in the first paragraph of the Discussion section and suggest that tailoring of physical activity interventions is needed to effectively reach rural cancer survivors.  This point is also emphasized in our abstract conclusion which relates directly to the main implementation outcome of limited program reach.  We did not specifically assess the underlying reasons for poor utilization in this study but suggest potential contributing factors in the Discussion section, paragraphs 3-5.  Additionally, we have added information in the Methods section to further describe the study design attributes requested by the reviewer.

Round 2

Reviewer 2 Report

Thanks for the changes I noted. I have no further requests for changes and I recommend the paper for publication.

Author Response

Thank you for your helpful comments.

This manuscript is a resubmission of an earlier submission. The following is a list of the peer review reports and author responses from that submission.

Round 1

Reviewer 1 Report

The aim of the manuscript titled ‘Implementation of Physical Activity Programs for Rural Cancer Survivors: Challenges and Opportunities’ is to evaluate the implementation of a clinic-based physical activity programme for cancer survivors at a rural community oncology setting.

The topic addressed in the manuscript is part of a wider theme, which is cancer, one of the most relevant problems from a public health perspective. Cancer is currently the second leading cause of death in the world and accounts for one in six of all deaths, totalising 9.6 million deaths a year. Within this context, cancer survival has been increasing. In the case of colon cancer, the 5-year net survival has increased from 58.5% in 1995 to 70.8% in 2014 in Australia, from 56.8% to 67.1% in Canada, and 44.4% to 59.0% in the United Kingdom. For lung cancer, the increase in the period 1995-2014 has been 12.6% to 21.3% in Australia, 14.9% to 22.6% in Canada and 6.86% to 14.7% in the United Kingdom. As it commonly occurs with all health indicators, there important variations among and within countries. One of the inequalities regularly reported is between urban and rural settings. In this sense, a research t that focuses on cancer survivors at a rural setting is a valuable effort.

The manuscript is both well organised and well written, which strengthen the communicational effectiveness of the paper. The title of the manuscript appropriately describes the content of the paper and the abstract reflects the content of the article. There is consistency among the different sections of the manuscript.

The Introduction section provides relevant information on evidence that would support the importance of physical activity to improve cancer–related health outcomes in cancer survivors, as well as inequalities of access to care for cancer patients living in rural settings and underlying factor that may explain these inequities. At the end of the Introduction section, the authors explicitly state the aim of the manuscript, which is appropriate since it contributes to modulate the reader’s expectations on the paper.

In order to achieve the study aim, the authors focused their study on cancer patients treated at Cowell Family Cancer Center/Munson Medical Center (CFCC/MMC), a rural regional cancer centre located in Northwest Michigan. Methodologically, the assessment of the implementation of the physical activity programme is based on the RE-AIM framework (Reach, Effectiveness, Adoption, Implementation, and Maintenance).

The authors do not explicitly state the study design. However, according to the information provided, they developed a quasi-experimental research with pre and post intervention (the clinic-based physical activity programme) assessment of the variables they were interested on: i) Reach of the physical activity programme (coverage of the physical activity programme among eligible cancer survivors treated at CFCC/MMC); ii) Effectiveness of the program (pre-post intervention change in quality of life of participants); iii) Fidelity to the physical activity program (proportion of cancer survivors participating in at least 5 physical activity classes). It is suggested that authors state the study design in order to facilitate the readers to understand the study rationale.

The result section is well organised into four sub-sections (study characteristics at baseline; program reach and fidelity; program effectiveness; changes in sleep) that are consistent with the variables of interest the authors described in the method section. The section presents findings in an adequate balance of text and both table 1 and figure 1, complementing each other and not being redundant.

The discussion is pertinent to the aim of the study. The authors provide a good analysis of their findings against the state of the art. The study findings are consistent with effectiveness of physical activity for improving cancer-related quality of life (including fatigue, constipation, and pain). It is also interesting the finding on benefits of physical activity on sleep quantity and quality among cancer survivors, which can be deemed as a contribution on an issue that still is subject to controversies.

The main finding of the study, from the personal perspective of this reviewer, is the extremely low coverage (reported as reach in the study) of the clinic-based physical activity programme that was assessed (only 0.59% of cancer survivors participating after an enrolment period of 29 months). This is the main foundation to advocate for physical activity programmes for cancer survivors that are specifically designed to serve people living in rural areas, such as home-based physical activity interventions the authors address in the discussion section of the manuscript.

No limitation of the study is reported by the authors in the paper. It is suggested that the authors do address limitations. One limitation is that patients of all the cancer types detailed in table 1 seem to have been exposed to the same physical activity programme. If this is true, it would be a relevant limitation since both medical and surgical treatments vary across cancer types (eg. Lung cancer vs multiple myeloma).

The conclusion section is consistent with the aim of the study and with all the information presented in the previous sections of the manuscript. In other words, the authors’ conclusion asserting that successful implementation of physical activity programs for cancer survivors will require tailoring to address specific needs and preferences of those persons has good foundations on the data the authors provide in the previous sections of the manuscript. In fact, providing adequate health care services (primary, secondary and tertiary prevention) to respond to needs and preferences of individuals and population is the permanent challenge for health care services.

Reviewer 2 Report

Please see the attached pdf file. 

Reviewer 3 Report

Re: Implementation of physical activity programs for rural cancer survivors: challenges and opportunities”

This study examined 24 cancer survivors who participated in the physical activity program during the period of May 2017 and September 2019 at Cowell Family Cancer Center/Munson Medical Center, a regional cancer center located in rural Northwest Michigan. The study claimed that significant improvements in cancer-related fatigue, constipation, pain, and sleep quality were observed after the program and suggested that physical activity interventions need tailoring to effectively reach rural cancer survivors.

I have to suggest rejecting the paper for publication on the basis of significantly flawed study design and methods, confusing presentations, as well as dubious conclusions.

Firstly, this study examined 24 cancer survivors from only one rural cancer center in a relatively short time period (around 2 years). Would these data be good in terms of quality and quantity enough to represent this population in general? Would the program have influences that gradually decrease or increase over time? Why did the authors only consider the effects after the program immediately? Additionally, findings of this study were obtained mainly from comparing patients’ self-reported data before and after the program. However, neither measures nor explanation were provided to address how subjective the data would affect the analysis results? Would other covariates, for example, sex, age, socio-economic status, education level, or environmental factors, would affect the results of both quality of life assessment and the implementation outcomes? Moreover, the study defined the reach of the physical activity program as the proportion of eligible cancer survivors treated at CFCC/MCC over the study period participating in the physical activity program, defined effectiveness as the change in QOL metrics before and after the participation in the program, and defined fidelity to the program by calculating the proportion of cancer survivor participating in at lease 5 physical activity classes. Neither references nor explanations were provided to address how and why they defined the three measures in these ways. To me, these definitions are too thin to describe the three dimensions of implementation outcomes: accessibility, effectiveness, and fidelity. For example, accessibility of the physical activity program could relate to patients’ age, sex, health status, and distance to cancer center, as well as how frequency in visiting the center. Moreover, statistical analysis methods adopted could not illustrate the effects before and after the program without adjusting for potential confounders of individual and contextual covariates. No sensitive analyses were conducted to address problems related to extreme data. Given the above, results generated from the above study design and analyses appeared to be unconvincing.